# Response to comment on 'Valid molecular dynamics simulations of human hemoglobin require a surprisingly large box size'

Krystel El Hage[1], Florent Hédin[1], Prashant K Gupta[1], Markus Meuwly[1]*, Martin Karplus[2,3]*

[1]Department of Chemistry, University of Basel, Basel, Switzerland; [2]Department of Chemistry and Chemical Biology, Harvard University, Cambridge, United States; [3]Laboratoire de Chimie Biophysique, ISIS, Université de Strasbourg, Strasbourg, France

**Abstract** We recently reported that molecular dynamics simulations for hemoglobin require a surprisingly large box size to stabilize the T(0) state relative to R(0), as observed in experiments (El Hage et al., 2018). Gapsys and de Groot have commented on this work but do not provide convincing evidence that the conclusions of El Hage et al., 2018 are incorrect. Here we respond to these concerns, argue that our original conclusions remain valid, and raise our own concerns about some of the results reported in the comment by Gapsys and de Groot that require clarification.
DOI: https://doi.org/10.7554/eLife.45318.001

*For correspondence:
m.meuwly@unibas.ch (MM);
marci@tammy.harvard.edu (MK)

Competing interests: The authors declare that no competing interests exist.

## Introduction

Our investigation of the box size dependence of hemoglobin in solution was initiated by the disagreement between the experimentally observed stabilization of the unliganded T state relative to an R-like structure (*Edelstein, 1971*), and molecular dynamics simulations which found that the unliganded T state (T(0)) was not stable and made a transition to an R-like state in times on the order of a hundred nanoseconds (*Hub et al., 2010*; *Yusuff et al., 2012*). We began research to determine what could be wrong with the published simulations and found, after looking at many possibilities, that only a surprisingly large (150 Å) simulation box was able to stabilize the T(0) over at least ~1.3 microseconds (*El Hage et al., 2018*; *Figure 1*).

## Results

### Statistics

To estimate rates, a sufficient number of samples is required. An explicit example, although not related to hemoglobin but illustrating the effect of insufficient statistics, is to consider reactive MD simulations for the vibrationally induced photodissociation of $HSO_3Cl$ to $SO_3$ and HCl (*Figure 2*). It is evident that with 20 events the distribution of reaction times is far from converged and does not even allow one to guess the true distribution: see *Yosa Reyes et al., 2016* for details on the simulations. This is also why from *ab initio* MD simulations of vibrationally induced photodissociation, no rates were determined because widely different reaction times for individual trajectories were found when only considering ~100 trajectories (see Figure 2 of *Miller and Gerber, 2006*). It also remains

unclear how the probability of 0.0026 was determined in *Gapsys and de Groot (2019)* (hereafter referred to as the comment).

## Hydrophobic effect

As stated in the abstract of the *El Hage et al. (2018)*: 'The results suggest that such a large box is required for the hydrophobic effect, which stabilizes the T(0) tetramer, to be manifested.' We felt we did not have conclusive evidence for the role of the hydrophobic effect, and have been working with Adam Willard to investigate this point. We note that the comment includes a list of some differences in the simulation setups: these and other differences in the two sets of setups need to be investigated as possible contributors to the difference in the simulation results. A possible problem with the comparisons in Figure 3 of the comment concerns the fact that we only exchanged input files for the 90 Å box; the authors of the comment constructed the larger boxes used in their simulations. In that regard, it is interesting to note that their observed life time for the 90 Å box with its error bars essentially includes the one observed in our simulation. Point on which we do not agree with the constructs used in the comment to reanalyze our results include the following:

\# The subvolume analysis which attempts to compare the diffusivity in large boxes but restricted to a smaller subvolume needs to introduce 'imaginary boundary conditions' and it is unclear how this was done. Such an approach appears to assume that water outside this artificial volume behaves as bulk water which is not evident. Alternatively, very short analysis times (~2 picoseconds) need to be used to avoid significant displacement of the water molecules (*Persson et al., 2017*).

\# The water-water hydrogen bond network reconfigures when the structure of the protein changes (T vs. R) and from one box size to another box size (see Figure 3 in *El Hage et al., 2018*). Following a transition in the protein structure these changes occur on the few nanosecond time scale, as a direct response to the protein structure change. This suggests that the protein and water dynamics are coupled and can not be viewed independently.

\# As an important test we have solvated the decayed (i.e., R-state) structure from the simulation in the 120 Å box in the 150 Å box and observed that the T-state is partially recovered (*Figure 3*). Apparently, solvation of the decayed structure in the larger water box provides the driving force required to partially recover the T(0) structure. This is an additional indication that, thermodynamically, T(0) is favored over R(0) in the largest water box without, however, providing information concerning a quantitative measure of the underlying energetics. Full recovery would require much longer simulations.

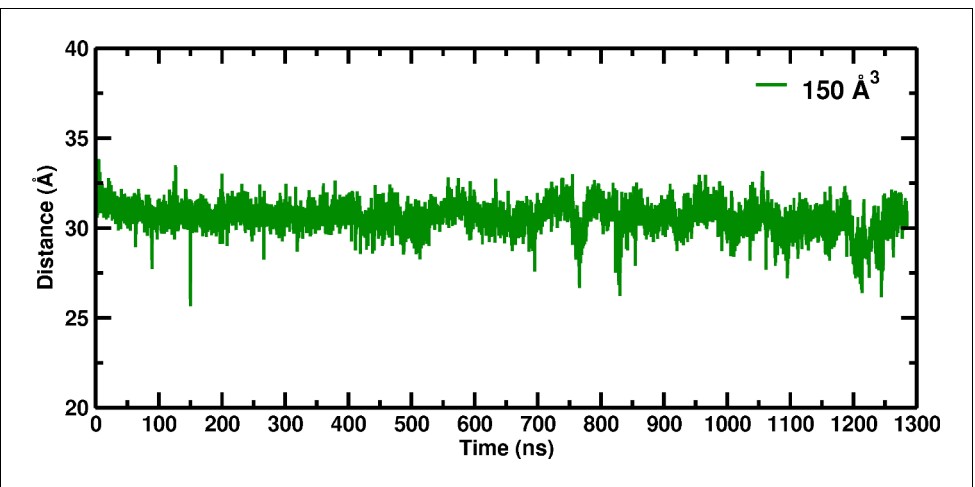

**Figure 1.** Temporal change of the Cα–Cα distance between $His_{146}B2$ and $His1_{46}B2$ for the 150 Å box during 1280 nanoseconds of molecular dynamics simulation.

DOI: https://doi.org/10.7554/eLife.45318.002

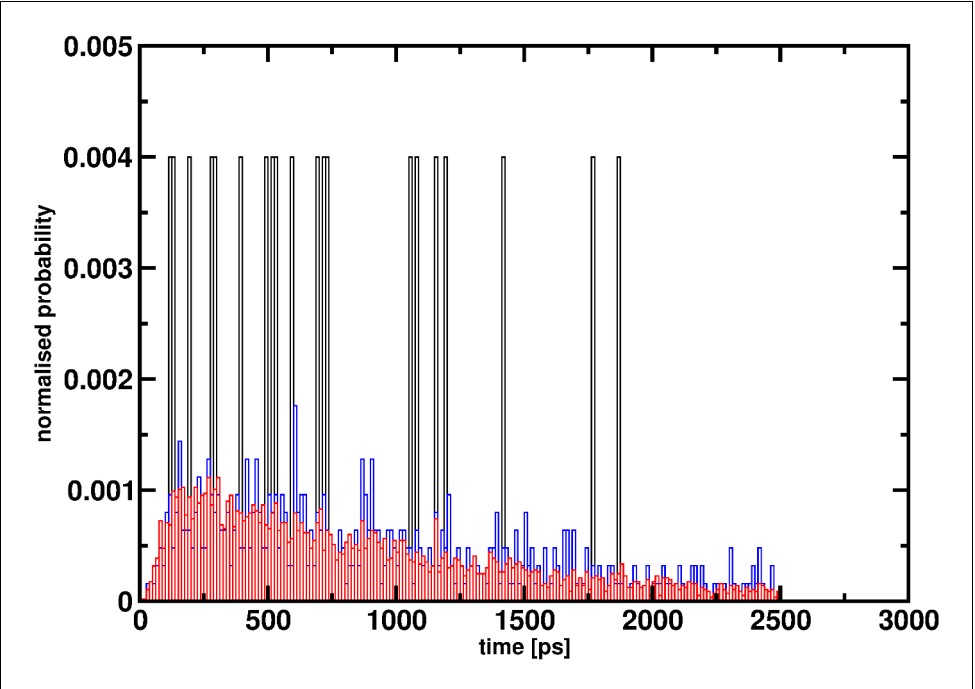

**Figure 2.** Reaction time distribution $p(\tau)$ for the decomposition $HSO_3Cl \to SO_3+HCl$ from 20 (black), 500 (red) and 5000 (blue) reactive trajectories. With 20 trajectories the distribution is far from converged and even with 500 trajectories convergence appears to be incomplete. It is only for 5000 independent events that $p(\tau)$ approaches a smooth distribution.

DOI: https://doi.org/10.7554/eLife.45318.003

# Technically, it is desirable to carry out simulations in a setup that allows the solvent – here water – to behave as a pure liquid for quantities such as $g(r)$ or the average number of H-bonds. With hemoglobin as the solute we find this to be the case for a box size of 150 Å.

The normalization procedure of the radial distribution functions used in the comment remains unclear. It should be noted that the radial distribution functions in *El Hage et al. (2018)* follow the expectations for spherical cavities (*Chandler, 2005*). Figure 5 in *El Hage et al. (2018)* aims at highlighting that in a box not sufficiently large, water barely reaches bulk properties as judged from the $g(r)$, whereas it readily does so when using a sufficiently large box (e.g. 150 Å). As no hydrophobic effect is expected for ubiquitin it is not clear what the data in Figure 2—figure supplement 1 of the comment contribute to relating the thermodynamic stability of the T(0) state of hemoglobin with the hydrophobic effect.

## Kinetics as a function of box size

This section raises questions as to the statistical significance of the results. As mentioned above and in *Figure 2*, converged reaction time distributions require hundreds to thousands of transitions. Given the fact that the T(0) state is unstable in the 90 Å and 120 Å boxes it is reasonable that starting from an ensemble of initial structures a distribution of decay times is found. Comparing two such distributions to provide evidence for the presence or absence of a box size dependence requires them to be converged to some degree which probably can not be achieved from 10 to 20 simulations.

It is stated in the comment that with the setup used in our simulations, 20 simulations were done for 90 Å and 150 Å boxes and 10 for a 120 Å box: however, the results shown in Figure 3 of the comment (that the transition is shorter in the larger boxes) is the inverse of what might be expected. This surprising result could be due to poor statistics, to some inconsistency in the use of our setup, or to something we do not understand. Interestingly, the T(0) lifetime in the 90 Å box, for which we had made input files available, is in the range we found (see Figure 1B in *El Hage et al., 2018*),

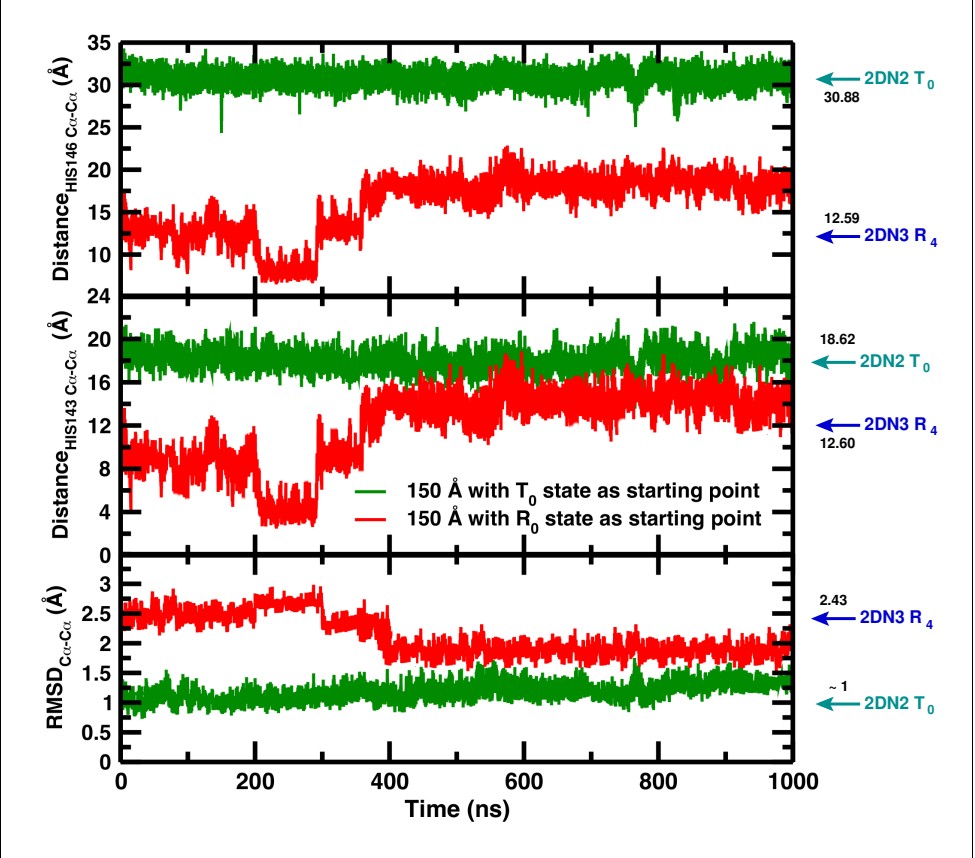

**Figure 3.** Temporal change in the 150 Å box of (from top to bottom) the Cα–Cα distance between His146B1 and His146B2, the Cα–Cα distance between His143B1 and His143B2, and the Cα RMSD relative to the 2DN2 X-ray structure. The green and red lines report the time series reported in our article (simulation starting from the T(0) state) and the time series of a simulation starting from the decayed T(0) state (i.e., an R(0) state structure) from R state in the 120 Å box, respectively. Cyan and blue arrows indicate the values of the corresponding observables found for the deoxy T(0)(2DN2) and oxy R(4)(2DN3) states, respectively.
DOI: https://doi.org/10.7554/eLife.45318.004

whereas this is not the case for the 120 Å and 150 Å boxes (for which we did not send input files). We also note a difference in the magnitude of the error bars in Figure 3A of the comment for the different protonation states which requires an explanation. Additional exchanges of information and detailed analyses would serve to clarify these points.

When we were still collaborating to investigate the origin of some of the results reported in the comment, we exchanged input files they used in their work and our input files. We note that a previous publication by one of the authors of the comment (*Hub et al., 2010*) does not provide details on the protonation states of the His, except for His146 for which they used either doubly protonated or protonation at ε. In another publication by one of the authors of the comment (*Vesper and de Groot, 2013*) it appears that the Kovalevsky protonation states that they sent us were not used. Thus we need more information to verify which protonation states were used in the various simulations. Our comparison showed that the protonation states of the histidine residues were very different (*Table 1*). We note that it is not sufficient to characterize the histidine residues as 'protonated' or 'deprotonated'. Rather, it is also relevant whether protonation is at the δ or ε positions, for example His143 which should be ε-protonated as it anchors the water network stabilizing His146 (see Figure 6 in *El Hage et al., 2018*).

Further differences between the simulations concern the shape of the simulation box (cubic in *El Hage et al., 2018* vs. dodecahedral), the center of mass of the protein was constrained to avoid periodic boundary effects (*El Hage et al., 2018*), and the terms for the dihedral parameters for the

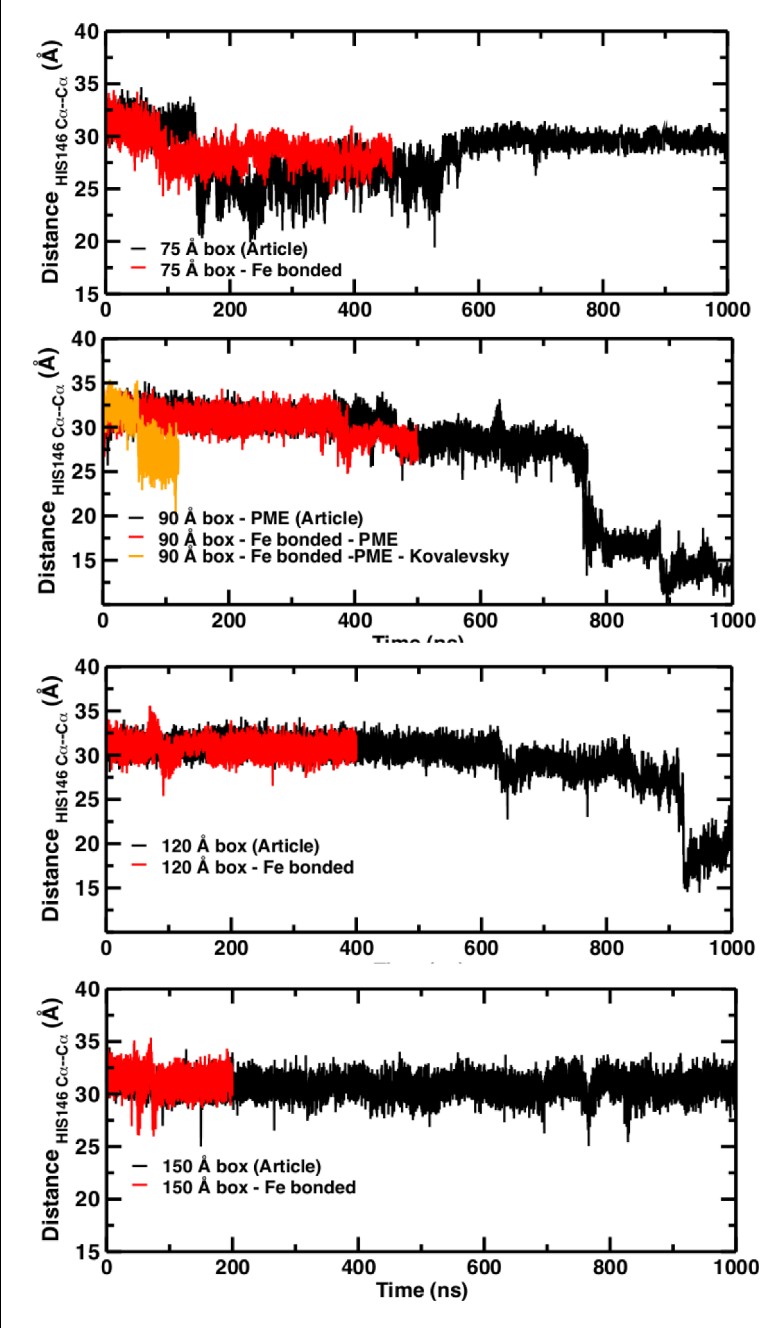

**Figure 4.** Temporal change of the Cα–Cα distance between His$_{146}$B1 and His$_{146}$B2 for the 75 Å box (**A**), 90 Å box (**B**), 120 Å box (**C**), and 150 Å box (**D**). In (**A**), (**C**), and (**D**) the black line reports the time series published and the red line shows the new time series obtained with the new simulation with the Fe–N bond. In (**B**) the black trace is from *El Hage et al. (2018)* the red trace for the bonded Fe–His simulations, the green line from using finite cutoffs for evaluating the electrostatics (cutoff 14 Å), and the orange line from using Kovalevsky protonation states for all histidines.

DOI: https://doi.org/10.7554/eLife.45318.006

Fe-proximal histidine differed. Removing the angular center of mass motion is only warned against if the motion of the center of mass itself is not controlled (i.e., if the solute can cross the boundaries of the periodic box). However, this is not the case in our simulations, but appeared to happen in some of those performed by the authors of the comment. Concerning the effect of the box shape on the

**Table 1.** Histidine protonation states in hemoglobin as used in simulations by the present authors (**Zheng et al., 2013**; **El Hage et al., 2018**), and the histidine protonation states in the files that were supplied to us by the authors of the comment (originally from **Kovalevsky et al., 2010**).

| Res. | Zheng et al., 2013; El Hage et al., 2018 Chain A/C | Kovalevsky et al., 2010 Chain A | Chain C | Res. | Zheng et al., 2013; El Hage et al., 2018 Chain B/D | Kovalevsky et al., 2010 Chain B | Chain D |
|---|---|---|---|---|---|---|---|
| 20 | HSE | HSP | HSE | 2 | HSE | HSE | HSE |
| 45 | HSE | HSE | HSE | 63 | HSE | HSP | HSE |
| 50 | HSD | HSD | HSP | 77 | HSE | HSE | HSE |
| 58 | HSE | HSP | HSE | 92 | HSD-Fe | HSD-Fe | HSD-Fe |
| 72 | HSD | HSP | HSP | 97 | HSE | HSP | HSP |
| 87 | HSD-Fe | HSD -Fe | HSD-Fe | 116 | HSE | HSP | HSP |
| 89 | HSE | HSE | HSP | 117 | HSE | HSE | HSE |
| 103 | HSE | HSP | HSP | 143 | HSE | HSE | HSP |
| 112 | HSE | HSP | HSP | 146 | HSP | HSP | HSP |
| 122 | HSE | HSE | HSE | | | | |

DOI: https://doi.org/10.7554/eLife.45318.005

proteins simulated under periodic boundary conditions, it has been found that the box type can have a statistically significant effect on the outcome of a simulation, and that the magnitude of the effect depends on the protein considered (**Wassenaar and Mark, 2006**). Although all of the latter may contribute to the difference in the results, we believe that a possible difference in the His protonation states used in the simulations would play a major role and needs to be examined.

## Kinetics rather than thermodynamics

A necessary but not sufficient criterion for thermodynamic stability is that the structure considered (here T(0)) does not decay on the time scale of the simulations (e.g., to R(0)). However, transition times from T to R can not be determined from such an approach. For this, either a very large number of independent simulations (on the order of hundreds to thousands; see **Figure 2**) are required, or free energy simulations need to be carried out. As pointed out in **El Hage et al. (2018)** from the available experimental data, the T(0) to R(0) transition time should be on the order of seconds. The transitions in simulation boxes of 75 Å, 90 Å, and 120 Å all were under one microsecond and increased with increasing box size (**Figure 4**). Panels (A), (B), and (D) show that the presence or absence of the Fe-His bond does not lead to substantial changes of the time series as far as they have been followed. Panel (B) also reports the changes for simulations with electrostatic cutoffs (green) and when simulations with the Kovalevsky protonation states were carried out (orange). With finite cutoff the T(0) state is stable for at least 150 nanoseconds, and with the Kovalevsky protonation it decays after 60 nanoseconds, in accord with earlier findings (**Hub et al., 2010**). The radial distribution functions around His146 in our original study (see Figure 6 in **El Hage et al., 2018**) and from our simulations using the Kovalevsky protonation in the 90 Å box suggest that with the latter protonation state His146 is typically overhydrated as compared with our (stable) simulation in the 150 Å box (see middle panel in **Figure 5**). Moreover, with the Kovalevsky protonation in the 90Å box, the height of the first and second peak of $g(r)$ is comparable to that of our own simulation in the 90 Å box after T(0) has decayed (at 500 nanoseconds and later; see **Figure 4B**).

Comparisons of simulations with and without a Fe–His bond show that the Fe–His bond lengths are comparable in all four subunits (see **Figure 6**). As to the Fe–Fe distances between the various heme groups, only the 120 Å and 150 Å box simulations gave results in agreement with experiment; the smaller boxes did not.

The 150 Å box in a T(0) state was also stable for at least 1.3 microseconds, the length of the simulation (**Figure 1**).

An obvious improvement of our initial study is to run multiple simulations for each box, in particular if one is interested in the kinetics of the T-to-R transition. We are doing additional simulations to examine the box size dependence of the lifetime. However, we point out that it is very difficult to do

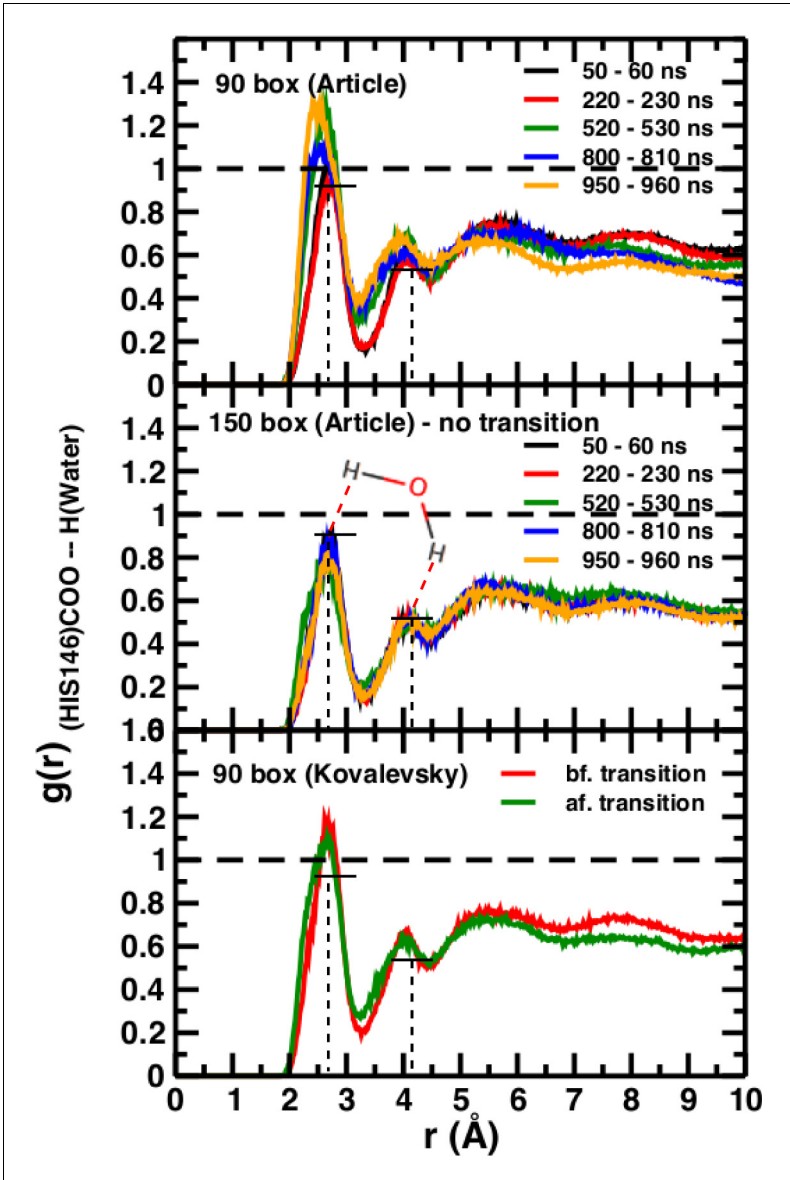

**Figure 5.** Time-averaged radial distribution functions *g(r)* between the C-terminal (COO) of His146 and water H for box sizes 90 Å (top) and 150 Å (middle) data from Figure 6 of *El Hage et al. (2018)* and from before and after the transition, obtained when using the Kovalevsky protonation in the 90 Å box (bottom).
DOI: https://doi.org/10.7554/eLife.45318.007

a sufficient number of independent simulations to obtain converged statistical results for a system the size of hemoglobin. In ten simulations for the 120 Å box reported in the comment, 4 out of the 10 lasted longer than one microsecond, compared with 3 out of 10 for the 90 Å box. In simulation of kinetics hundreds to thousands of individual, reactive trajectories are required for converged reaction time distributions that can be compared with experiment (*Yosa and Meuwly, 2011*; *Soloviov et al., 2016*). The data in Figure 3 of the comment cannot report on the kinetics because the number of events observed is too small (*Figure 2*).

## Kinetics as a function of box size

We refer readers to our author response. We also note that previous work (*Wassenaar and Mark, 2006*) did find a dependence on box shape.

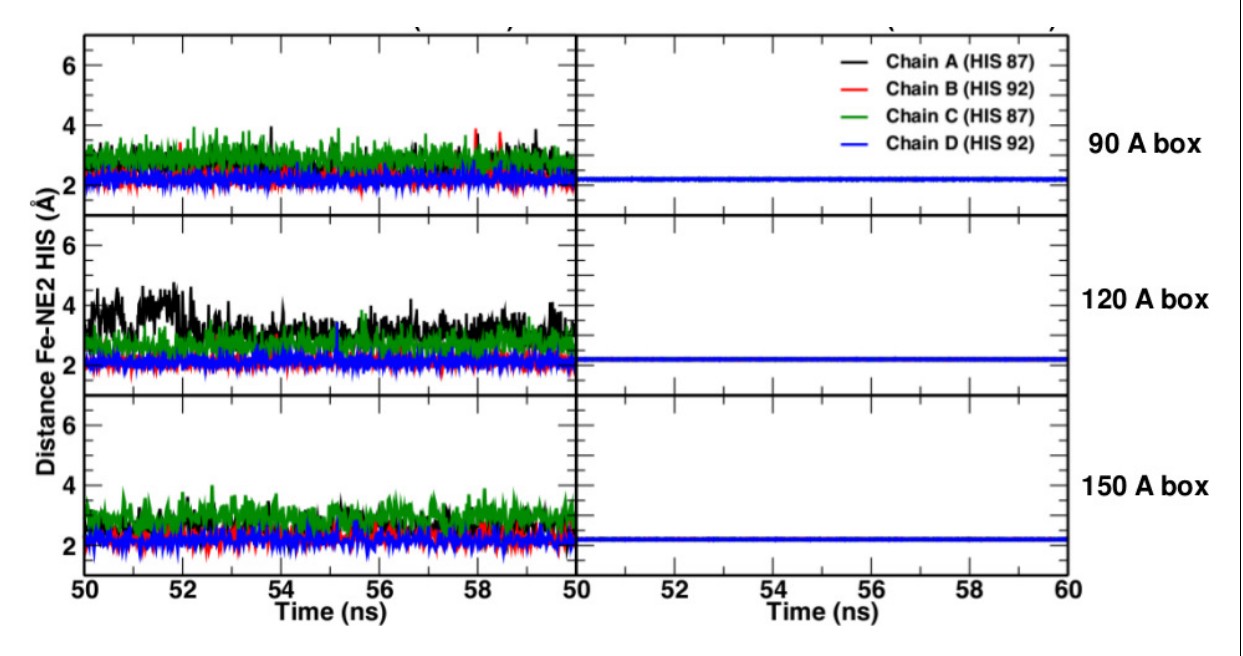

**Figure 6.** Fe-N$_{His}$ bond lengths in simulations without (left) and with (right) explicit bond.
DOI: https://doi.org/10.7554/eLife.45318.008

## Thermodynamics as a function of box size

The results for the other two systems are interesting but there is no reason to expect a box size dependence for the structural changes that were studied. Among other points, the surface area change is small and a hydrophobic effect is unlikely. There is no direct relationship to our findings for hemoglobin.

## Conclusions

We believe we have demonstrated that the details of our simulations provide evidence that there is a box size dependence in hemoglobin simulations and that it is likely, though not proved, that in a 150 Å box the unliganded T state is stable. The role of the particular choice of His protonation states in the fast decay presented in the comment remains to be evaluated. Finally, we reiterate that the unliganded T state is known to be stable from experiment: therefore, if our results were not correct, there would have to be a yet unknown problem with simulations of unliganded T state hemoglobin.

## Additional information

### Funding

| Funder | Grant reference number | Author |
|---|---|---|
| Swiss National Science Foundation | 200021-117810 | Markus Meuwly |
| Swiss National Science Foundation | NCCR MUST | Markus Meuwly |
| CHARMM Development Project | | Martin Karplus |

The funders had no role in study design, data collection and interpretation, or the decision to submit the work for publication.

## Author contributions
Krystel El Hage, Investigation, Methodology, Writing—original draft, Writing—review and editing; Florent Hédin, Prashant K Gupta, Writing—review and editing; Markus Meuwly, Conceptualization, Resources, Formal analysis, Supervision, Validation, Methodology, Writing—original draft, Writing—review and editing; Martin Karplus, Writing—original draft, Writing—review and editing

## Author ORCIDs
Krystel El Hage (iD) http://orcid.org/0000-0003-4837-3888
Florent Hédin (iD) https://orcid.org/0000-0001-6341-7557
Prashant K Gupta (iD) http://orcid.org/0000-0002-4792-7538
Markus Meuwly (iD) https://orcid.org/0000-0001-7930-8806

## Decision letter and Author response
Decision letter https://doi.org/10.7554/eLife.45318.011
Author response https://doi.org/10.7554/eLife.45318.012

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
