## [Decision Letter]

[Editors’ note: the original decision letter was sent on 20 March 2019, and was followed by further letters on 29 April 2019 and 7 May 2019: the point-by-point response by Gapsys and de Groot was attached to the last of these letters.]

20 March 2019:

Thank you for submitting your response to the comment by Gapsys and de Groot on your eLife article "Valid molecular dynamics simulations of human hemoglobin require a surprisingly large box size". Your response has been reviewed by two peer reviewers, and the evaluation has been overseen by a Reviewing Editor (Yibing Shan), a Senior Editor (John Kuriyan) and the eLife Features Editor (Peter Rodgers).

We plan to progress this matter as follows:

- i) We will ask Gapsys and de Groot to revise their comment to address the points made by the referees (which include a request to delete some material) and to clarify some of the points raised in your response to their comment.

- ii) The comments of the referees on your response are below. No action with respect to these comments is needed at this stage.

- iii) Once we have received the revised version of the comment from Gapsys and de Groot, we will send it to you so that you can revise your response accordingly. At this stage we will also request some editorial revisions and ask for a point-by-point response to the points in the decision letter (as is our standard practice).

29 April 2019:

Further to our email of March 20 (above), we have now received the revised version of the comment by Gapsys and de Groot, so we are writing with the details of the revisions that you need to make to your response to this comment. As indicated in the email of March 20, you will need to make: i) editorial changes; ii) changes in response to the reviewer comments; iii) changes to reflect the changes in the comment by Gapsys and de Groot. Please see below for more details.

Please be aware that if your manuscript is accepted for publication, we will publish the decision letter and your point-by-point response to it (excluding minor comments).

# Editorial changes

a) Please change the title to the following (which is the standard format for such articles in eLife):

Response to comment on 'Valid molecular dynamics simulations of human hemoglobin require a surprisingly large box size'

b) Please revise the abstract so that it better summarizes your manuscript. I would suggest something along the following lines:

"In a previous paper we reported that [QUERY: Please summarize your original paper and the dependence on box size here] (El Hage et al., 2018). Gapsys and de Groot have expressed a number of concerns about this work, and have argued that the dependence on box size that we reported is not reproducible (Gapsys and de Groot, 2019). Here we respond to these concerns and argue that our original conclusions remain valid."

Please feel free to revise the above, but please avoid stating that Gapsys and de Groot are incorrect. Please also answer the query.

c) Please revise the conclusion so that, again, it better summarizes your manuscript. I would suggest something along the following lines:

"We continue to believe that our work provides evidence for a dependence on box size in molecular dynamics simulations of human hemoglobin, and that is likely, though not proven, that the unliganded T state of human hemoglobin is stable in a 150 Å box. We also believe that the role of the His protonation states in the fast decay of the T(0) state [OK?] reported by Gapsys and de Groot remains to be evaluated. Finally, we reiterate that the unliganded T state is known to be stable from experiment: therefore, if our results are not correct, there exists an as yet unknown problem with the simulations of unliganded T state hemoglobin."

Please feel free to revise the above.

# Changes in response to the reviewer comments

The reviewer comments (which were included in the email of 20 March) are copied below. Of the six points made by reviewer #1, points 1 and 5 are covered below (under the heading "Changes to reflect the changes in the comment by Gapsys and de Groot"), and points 2, 3 and 4 do not require a response. The four paragraphs from reviewer #2 do not require a response.

Reviewer #1:

This paper is to reply to the comment by Gapsys and de Groot on the published work "'Valid molecular dynamics simulations of human hemoglobin require a surprisingly large box size' by El Hage et al. The reply was focused on that (a) the examination work does not invalidate the original conclusion that there is a significant box size effect and (b) several statements in the comment need to be clarified. With careful review of both the comment by Gapsys and de Groot and this reply, by and large, I agree with the general discussion of El Hage et al., particularly on the fact that the observed box size dependency is interesting and needs to be further explored; and it is likely an explanation to the mystery of unexpected fast T-R transition in MD simulations. Given that, I have the following opinions and suggestions:

1) I have suggested some major changes to Gapsys and de Groot regarding their focus of the discussion. Hopefully they will revise the comment accordingly and the authors of this reply can change their reply accordingly.

2) I appreciate the fact that El Hage et al. further clarified the fact that the hydrophobic effect was proposed as an hypothesis rather than a firm conclusion and they are further exploring this effect.

3) I appreciate the fact that El Hage et al. further clarified that the focus of the original paper is to "demonstrate that the details of the simulations provide evidence that there is a box size dependence in hemoglobin simulations and that is likely, though not proved, that in a 150 Å box the unliganded T state is stable."

4) In my opinion, the importance of the original work even with the above two clarifications still firmly holds and the paper will be an important literature in the field of MD simulation.

5) I agree on the proposal by El Hage et al. that to re-exam their original observation, the simulation details need to be carefully checked. I would like to encourage the authors of both sides to work together to further explore these important observations in the near future.

6) A minor suggestion is that El Hage et al. may consider putting some appreciation comments on the fact that their work has attracted attentions of the authors of the comment; and Gapsys and de Groot have put quite a bit efforts in further looking to this issue and provided some suggestion on some aspect of the analysis. It is a healthy scientific process.

Reviewer #2:

While it is an interesting and potentially important possibility that simulation box size may affect simulation outcome, I think it is clear that the statistics in the paper by El Hage et al. are insufficient to establish this. In the comment by Gapsys and de Groot, more repeats of the simulation of hemoglobin were reported, but the number of the repeats is not large enough to exclude this possibility, although the simulations appear to be inconsistent with this possibility. The suggested effect of simulation box size is necessarily a subtle one, if it does exist, and the value of the comment, in my assessment, is in pointing out that this is an unresolved issue that require further investigations. This, importantly, is acknowledged by El Hage et al in their reply to the comment.

Gapsys and de Groot correctly pointed out that even the transition time is established to be smaller in simulations of a small box than in ones of a big box, one cannot reliably infer that the thermostability of the initial T state is reduces by the small box size. However, this is common knowledge of even beginners of this field, of which surely El Hage et al. are aware. This distinction between kinetics and thermodynamics is peripheral at this point when it is not known for certain whether the transition time is indeed affected by box size. I think this section should be removed from the comment.

Gapsys and de Groot proposed an alternative approach to analyze the potential hydrophobic effect associated with box size and showed that such an approach would lead to a different conclusion. This is valid. Neutral statements to this effect would suffice.

The suggested effect of simulation box size is necessarily a subtle one. Thus the fact that Gapsys and de Groot did not see it in simulations of a number of other systems is not sufficient to exclude this possibility. It seems unnecessary to report these simulations in the comment. Or at least, it should be acknowledged that these additional simulations are not conclusive either way.

# Changes to reflect changes in the Comment by Gapsys and de Groot

Please address the following points by revising your manuscript and explaining in your point-by-point response how your manuscript has been revised. It is important that you make these changes so that readers of the two articles can get a clear idea of what has been done by both groups.

i) The order of subsections within the Results section of the comment by Gapsys and de Groot has been changed, so please change the order of the subsections within the Results section of your manuscript.

ii) Gapsys and de Groot have expanded the "Statistics" subsection by revising the sentences that start "For example, we find.. . " and "This is further underscored... " They have also added eight new sentences to the end of this section, and have added a new figure (Figure 1-figure supplement 1).

Please consider revising your manuscript to reflect these additions.

iii) Gapsys and de Groot have expanded the "Hydrophobic effect" subsection by adding six new sentences to the end of this sectio. They have also added a new figure (Figure 2-figure supplement 1).

Please consider revising your manuscript to reflect these additions.

iv) Gapsys and de Groot have added a new sentence to the end of the "Kinetics rather than thermodynamics" subsection.

Please consider revising your manuscript to reflect this addition.

v) We asked Gapsys and de Groot to revise the previous version of this manuscript to better explain how their setups compared to/differed from your setups. Our requests to them are in regular type below; their replies to these requests are in italic type. Please revise your manuscript as necessary in the light of these changes:

"We note that the setups by El Hage et al and by Gapsys and de Groot are not identical, and the differences in the setups potentially involve several parameters (e.g., how the initial water box was constructed, how long the water molecules were equilibrated before the production runs, whether the protein system was restrained during the water equilibriation, and what types of baro-stat and thermo-stat were used et al.) To clarify these issues, please be explicit about the differences between your setups and the setups used by El Hage et al.

REPLY: We are aware of this issue and therefore had contacted El Hage et al for their setup. The data in our comment that refer to 'El Hage setup' is based on their 90A setup and therefore is identical to that of El Hage et al in all the mentioned aspects of how the initial water box was constructed, how long the water molecules were equilibrated before the production runs, how the protein system was restrained during the water equilibriation, what force field was applied, and what types of barostat and thermostat et al were used. Therefore, the reviewers' assertion above "We note that the setups by El Hage et al and by Gapsys and de Groot are not identical" appears incorrect. Based on this setup of the 90A box, we ran 20 simulations of 1 microsecond each. 9 of these did not complete the transition, and thus remain in the T state, which in the El Hage et al. 2018 study had been claimed as an exclusive feature of the 150A box. The fact that we observe the lack of a transition also in equally long simulations in the 90A box thus renders the conclusion of El Hage et al. invalid that the lack of a transition to R (or indeed a valid description of deoxyhemoglobin to remain 'stable' in T) is due to the larger box size in the 150A box.

In addition, we have also carried out simulations in larger boxes of cubic sizes of 120A and 150A. The 120A and 150A boxes were constructed from the 90A box by adding a water layer (with solvated ions) of 30 and 60A, respectively. These therefore also are identical in terms of protein configuration as well as protein solvation shell to the El Hage et al setup. We have now added this description to the manuscript. That the added solvation has created a setup indistinguishable from the El Hage et al setup for the 120A and 150A boxes is illustrated by the fact that the single trajectory findings of El Hage et al. fall well within the error bars that we provide not only for the 90A box but also for the 120A and 150A boxes. This can be even more clearly seen in our Fig. 1B, that shows that the El Hage et al. simulations fall well within the distributions we find for all the investigated box sizes.

That other parameters in the setup indeed have an influence on the transition statistics is reflected in Fig. 3A where we show that three different hemoglobin setups in terms of e.g. chosen protonation states or protein-heme interactions show three different types of transition kinetics. In none of these three, we see a systematic effect of the box size, however.

Also, please revise your manuscript to address the following comments in the response from El Hage et al. a) The comments in the sentence that starts "A possible problem with the comparisons in their figure 3…"

REPLY: The setup issue is already addressed above. It was the choice of El Hage et al to only send us their 90A setup (we had before that shared all our setups with them). The only thing we did to construct the 120A and 150A boxes was to add a layer of solvent with ions around the pre-solvated 90A box. In the same paragraph, El Hage et al write: "In that regard, it is interesting to note that their observed life time for the 90A box with its error bars essentially includes the one observed in our simulation." The same is in fact true for the 120A and 150A boxes: the single trajectory findings of El Hage et al. fall well within the error bars that we provide. This can be even more clearly seen in our Fig. 1B, that shows that the El Hage et al. simulations fall well within the distributions we find. There is thus no reason to assume 'A possible problem' with the 120A and 150A boxes.

b) The comments in the sentence that starts "We also note a difference in the magnitude of the error bars.. . "

REPLY: the errors are different because the transition times are different. Smaller absolute transition times result in smaller associated absolute errors. We now included this notion in the revised manuscript.

c) The comments in the paragraph that starts "When we were still collaborating…"

REPLY: This section is about the effect of protonation states. For the systems labeled 'El Hage' in our comment, the protonation states are identical to the ones of El Hage et al (2018) as the simulations started from the input provided by El Hage et al. In addition, as pointed out in our comment, The El Hage et al protonation states are nearly identical to the ones in our "Hub" setup (except for one small difference of a surface histidine of which we argue that it is unlikely to substantially affect the transition kinetics) and therefore it seems odd to state (like El Hage et al do): "protonation states of the histidine residues were very different". They are different to the Kovalevsky et al. setup, the effect of which on hemoglobin we had already reported in 2013 (Vesper and de Groot, 2013). This section in the reply of El Hage et al thus seems rather confusing and does not appear to provide novel insight. In addition, the protonation states listed in Table 1 of the El Hage et al reply differ from the setup that was sent to us.

In total, we tested 3 different hemoglobin setups with different protonation states and observed no box size dependence in any of them."

7 May 2019:

Thank you for your email. I attach the point-by-point response from Gapsys and de Groot in response to the referee reports on the previous version of their manuscript. Please treat this document in confidence (and please do not quote from it in your revised manuscript).

As requested in my email of April 29, can you please revise your manuscript to i) address the editorial points outlined in that letter; ii) address the comments from the referees included in that letter; iii) to reflect the changes in the manuscript of Gapsys and De Groot. Please note that it should be possible to address all these points by making textual revisions to your manuscript: that is, we are not asking you to perform further simulation.

You will be able to use the following link to upload your revised manuscript and your point-by-point response to the email of April 29:

I look forward to receiving your revised manuscript and point-by-point response. It would be helpful if you could submit these within the next two weeks (i.e., by Tuesday 21 May). If you think you will need more time to revise your manuscript, please let me know.

Finally, please note that eLife publishes the decision letter and the point-by-point response it (excluding minor comments) for all papers.

---

## [Author Response]

In response to the comments in the decision letter we have made the following changes to our Response to the Comment from Gapsys and de Groot.

# Editorial changesa) Please change the title to the following (which is the standard format for such articles in eLife): Response to comment on ‘Valid molecular dynamics simulations of human aemoglobin require a surprisingly large box size’

REPLY: This was done.

b) Please revise the abstract so that it better summarizes your manuscript. I would suggest something along the following lines: “In a previous paper we reported that [QUERY: Please summarize your original paper and the dependence on box size here] (El Hage et al., 2018). Gapsys and de Groot have expressed a number of concerns about this work, and have argued that the dependence on box size that we reported is not reproducible (Gapsys and de Groot, 2019). Here we respond to these concerns and argue that our original conclusions remain valid." Please feel free to revise the above, but please avoid stating that Gapsys and de Groot are incorrect. Please also answer the query.

REPLY: The abstract was partly rewritten. We did not state that Gapsys and de Groot are incorrect, hence, no correction was required.

c) Please revise the conclusion so that, again, it better summarizes your manuscript. I would suggest something along the following lines: “We continue to believe that our work provides evidence for a dependence on box size in molecular dynamics simulations of human hemoglobin, and that is likely, though not proven, that the unliganded T state of human hemoglobin is stable in a 150 box. We also believe that the role of the His protonation states in the fast decay of the T(0) state [OK?] reported by Gapsys and de Groot remains to be evaluated. Finally, we reiterate that the unliganded T state is known to be stable from experiment: therefore, if our results are not correct, there exists an as yet unknown problem with the simulations of unliganded T state hemoglobin.” Please feel free to revise the above.

REPLY: The Conclusion was partly rewritten.

# Changes in response to the reviewer comments

Reviewer 1:

1) I have suggested some major changes to Gapsys and de Groot regarding their focus of the discussion. Hopefully they will revise the comment accordingly and the authors of this reply can change their reply accordingly.REPLY: Apparently the reviewer had concerns about the Comment by Gapsys and de Groot, which were not addressed since no changes were made by Gapsys and de Groot in their Comment, other than to report additional simulations.

REPLY: Please see below for our comments on the point-by-point response from Gapsys and de Groot.

5) I agree on the proposal by El Hage et al. that to re-exam their original observation, the simulation details need to be carefully checked. I would like to encourage the authors of both sides to work together to further explore these important observations in the near future.

REPLY: Please see for our comments on the point-by-point response from Gapsys and de Groot (below).

Reviewer 2: no response required

# Changes to reflect changes in the Comment by Gapsys and de Grooti) The order of subsections within the Results section of the comment by Gapsys and de Groot has been changed, so please change the order of the subsections within the Results section of your manuscript.

REPLY: This was done.

ii) Gapsys and de Groot have expanded the “Statistics” subsection by revising the sentences that start “For example, we find.. . ” and “This is further underscored... “. They have also added eight new sentences to the end of this section, and have added a new figure (Figure 1-figure supplement 1). Please consider revising your manuscript to reflect these additions.

REPLY: We expanded our own comments on this point and provide quantitative information that from 10 or 20 reactive events the kinetics of a process cannot be reliably described.

iii) Gapsys and de Groot have expanded the “Hydrophobic effect” subsection by adding six new sentences to the end of this section. They have also added a new figure (Figure 2-figure supplement 1). Please consider revising your manuscript to reflect these additions.

REPLY: We included our reservations towards the normalization procedure which is still not satisfactorily explained. We also comment directly on the new Figure 2-figure supplement 1.

iv) Gapsys and de Groot have added a new sentence to the end of the “Kinetics rather than thermodynamics” subsection. Please consider revising your manuscript to reflect this addition.

REPLY: As explained in the 'Statistics' subsection, the number of events (10 to 20) is too small to provide reliable information about the kinetics, see also our new Figure 2. We added a comment on that to the 'Kinetics rather than thermodynamics' subsection.

v) We asked Gapsys and de Groot to revise the previous version of this manuscript to better explain how their setups compared to/differed from your setups. Our requests to them are in regular type below; their replies to these requests are in italic type. Please revise your manuscript as necessary in the light of these changes.

REPLY: No change was made in our Response to the comment. See also for our comments on the point-by-point response from Gapsys and de Groot (below)

We thus have addressed all points raised by the reviewers and we hope that the work can now be accepted for publication.

[Editors’ note: in the rest of this document, Profs Muewly and Karplus respond to the author response from Gapsys and de Groot.]Point 1.

In point 1, the reviewer focuses on the possibility that the box size effect on the conformational transition of large proteins cannot be excluded, although it is far from established by the work of El Hage et al, 2018. The Comment from Gapsys and de Groot reports more repeats of simulations (approximately 10), than El Hage et al. (just one), but it also falls short of establishing the statistical significance of an opposite observation for which hundreds or more simulations would be required. We have provided concrete data for a much smaller system in Figure 2 in our Response to the Comment (El Hage et al., 2019).

"The reviewers believe it is important for both parties to not overstate the case, and to clearly acknowledge that the box size effect remains a hypothesis, neither established nor convincingly precluded. Unless the authors think this is not the case, the comment should be revised to reflect this basic reality that neither parties in this debate knows the ultimate answer here."

In response to this comment of the reviewers, Gapsys and de Groot introduce a comment about the very low probability of longer lifetimes but present no justification for this statement. Also, they quote a paper, which claims that only 10 repeats are required to obtain a statistically significant result. This contrasts sharply with several published papers (Yosa and Meuwly, 2011; Yosa Reyes et al., 2016; see also Figure 2 in El Hage et al., 2019) where thousands of events were found to be required to describe the kinetics. For a comparison, see Figure 2 in Miller and Gerber, 2006, where 100 simulations were not sufficient to converge the reaction probability.

Point 2.

The reviewer states "The discussion on the distinction between thermal and kinetic stability in the first paragraph of the results section is not needed and should be removed. (The corresponding section in the reply by El Hage et al. should be also be removed.)"

Here we agree with Gapsys and de Groot that the question of whether a kinetic or thermodynamic effect is involved is important and should be discussed. However, this should only be addressed once the simulation conditions have been established that lead to a thermodynamically stable T(0) state, consistent with experiment. Moreover, we disagree with the rationale of Gapsys and de Groot for doing so. The purpose of El Hage et al. (2018) was never to quantify the actual kinetics (i.e. transition time). In fact, it is stated in El Hage et al., 2018 that "..the R(0) to T(0) transition time is about 20 $\mu$s, much longer than the simulation time."

El Hage et al., 2018 reported on a possible simulation box size effect on the T(0) stability and was motivated, as mentioned at the beginning of this reply, by several published simulations of the T(0) state of hemoglobin, of which Hub et al., 2010) was one. As stated in the Introduction of El Hage et al., 2018, it is known from experiment that T(0) is stable so that the observation in the simulations of Hub et al., 2010 that the T(0) state goes over to an R(0)-like state indicates that there is something wrong with the thermodynamics. Further, that the T(0) to R(0) transition takes place on the hundred nanosecond time scale, while it is known from experiment (see El Hage et al., 2018) that such rare transitions require seconds, indicates that something is also wrong with the kinetics in the simulations of Gapsys and de Groot.

Point 3.

The reviewer begins with the statement "It is known though rarely formally acknowledged, that when the simulation length is far shorter than the actual time scale of a molecular event, the simulation results are sensitive to the exact simulation set up. We note that the setup of El Hage and by Gapsys and de Groot are not identical and the differences in the setup potentially involve several parameters."

The reviewer then lists several differences between the simulation setups.

In their response, Gapsys and de Groot point out that for the 90Å box simulated with Gromacs we had sent them our input files (as we believed we were still collaborating), and so it is likely that the simulation conditions were identical. However, for the 120Å and 150Å boxes they did not have our input files and so they generated the larger boxes in a rather arbitrary manner by "adding a water layer (with solvated ions of 30 Angstroms and 60 Angstroms, respectively." There is no reason to presume that this way of creating the larger boxes yielded a setup that is identical to the ones we used. Given the coupling between the protein conformation and water (pointed out on page 4 of the Reply to the Comment and elaborated under point 4), it is very likely that the setups were, in fact, significantly different.

We also note here that Gapsys and de Groot list a set of differences between our simulations and theirs that could be important since the effect is subtle. Reviewer 1 suggests further communication to resolve the difference. Gapsys and de Groot declined to do so.

As to specific points of the reviewer where Gapsys and de Groot are asked to revise their manuscript

- point (a) See comments above about the 120Å and 150Å boxes.

- point (b) the response of Gapsys and de Groot is appropriate.

- point (c) See the comments about the protonation states used in the simulations in the section 'Kinetics as a function of box size' in the main article.

Point 4.

This comment refers to the hydrophobic effect and the reviewers "suggest that in the revision the authors make clear that the hydrophobic effect hypothesis is not precluded although the original analysis by El Hage can be improved." Gapsys and de Groot argue that the various effects observed by El Hage, which suggested that there is a box size dependence, can be explained as a simple dilution effect. However, as stated in our Reply, "we do not agree with the constructs used in the Comment to reanalyze our results. These include: The subvolume analysis which attempts to compare the diffusivity in large boxes but restricted to a smaller subvolume needs to introduce 'imaginary boundary conditions' and it is unclear how this was done. Such an approach also appears to assume that water outside this artificial volume behaves as bulk water. Alternatively, very short analysis times (~2 ps) need to be used to avoid significant displacement of the water molecules (Persson et al., 2017)."

The water-water hydrogen bond network reconfigures when the structure of the protein changes (T vs. R) and from one box size to another box size (see Figure 3 in El Hage et al., 2018). Following a transition in the protein structure these changes occur on the few nanosecond time scale, as a direct response to the protein structure change. This suggests that the protein and water dynamics are coupled and can not be viewed independently."

This second point is an essential element that was neglected in the constructs of Gapsys and de Groot and makes clear that the observed behavior cannot be ascribed to a simple dilution effect. Gapsys and de Groot responded: "...box size effect on the hydrophobic effect (as quantified by hydrogen bonding, solvent RDF..) disappears when normalized.." This basically says - or at least implies - that the analysis in Chandler, 2005 is incorrect as this work links the hydrophobic effect to the behaviour of the RDF and our analysis for the different box sizes reflect exactly that behaviour of the RDF. So something with the "normalization" in Gapsys and de Groot must be wrong.

Point 5.

The reviewer points out that "The authors' simulations of small proteins without observing a box size effect does not serve as proof to preclude the box size effect, especially for large systems. The authors should make this important caveat clear in their discussion." In fact, reviewer 2 in his reply to us states "… it seems unnecessary to report these simulations in the Comment. Or at least … are not conclusive either way." Gapsys and de Groot state that they limit their findings to "for the investigated systems". Thus, they do mention the limitation, but the emphasis of what they wrote and what the reviewer requests are different.